# Prevalence and risk factors for latent tuberculosis infection among household contacts of index cases in two South African provinces: Analysis of baseline data from a cluster-randomised trial

**Peter MacPherson**[1,2]*, **Limakatso Lebina**[3], **Kegaugetswe Motsomi**[3], **Zama Bosch**[3], **Minja Milovanovic**[3], **Andrew Ratsela**[4], **Sanjay Lala**[5], **Ebrahim Variava**[6], **Jonathan E. Golub**[7], **Emily L. Webb**[8], **Neil A. Martinson**[5,7]

1 Department of Clinical Sciences, Liverpool School of Tropical Medicine, Liverpool, England, United Kingdom, 2 Malawi-Liverpool-Wellcome Trust Clinical Research Programme, Blantyre, Malawi, 3 SA MRC Soweto Matlosana Collaborating Centre for HIV/AIDS and TB, Perinatal HIV Research Unit, University of the Witwatersrand, Johannesburg, South Africa, 4 Department of Internal Medicine, University of Limpopo, and Limpopo Department of Health, Polokwane, South Africa, 5 Department of Paediatrics and Child Health, Chris Hani Baragwanath Academic Hospital and University of the Witwatersrand, Johannesburg, South Africa, 6 Department of Internal Medicine, Klerksdorp Tshepong Hospital Complex, North West Department of Health and University of the Witwatersrand, Polokwane, South Africa, 7 Johns Hopkins University Center for TB Research, Baltimore, Maryland, United States of America, 8 MRC Tropical Epidemiology Group, London School of Hygiene and Tropical Medicine, London, England, United Kingdom

* peter.macpherson@lstmed.ac.uk

## Abstract

### Introduction

Household contacts of patients with active pulmonary tuberculosis (TB) often have latent TB infection, and are at risk of progression to disease. We set out to investigate whether index TB case HIV status was linked to a higher probability of latent TB infection among household contacts.

### Materials and methods

Data were collected prospectively from participants in the intervention arm of a household cluster-randomised trial in two South Africa provinces (Mangaung, Free State, and Capricorn, Limpopo). In intervention group households, TB contacts underwent HIV testing and tuberculin skin testing (TST). TST induration was estimated at two cut-offs ($\geq$5mm, $\geq$10mm). Multilevel Bayesian regression models estimated posterior distributions of the percentage of household contacts with TST induration $\geq$5mm and $\geq$10mm by age group, and compared the odds of latent TB infection by key risk factors including HIV status index case age and study province.

### Results

A total of 2,985 household contacts of 924 index cases were assessed, with most 2,725 (91.3%) undergoing TST. HIV prevalence in household contacts was 14% and 10% in

**Data Availability Statement:** Code and datasets to reproduce analysis are available to download from https://github.com/petermacp/tstsa

**Funding:** This study was funded by the South African Medical Research Council and the UK MRC Newton Fund (SAMRC grant number: 96783). The South African Government funded all laboratory investigations. Support and approval was obtained from the Provincial Departments of Health of the Limpopo and Free State Provinces. PM is supported by Wellcome (grant number: WT089673). The funders had no role in the analysis, writing or decision to submit the manuscript.

**Competing interests:** The authors have declared that no competing interests exist

Mangaung and Capricorn respectively. Overall, 16.8% (458/2,725) had TST induration of $\geq$5mm and 13.1% (359/2,725) $\geq$10mm. In Mangaung, children aged 0–4 years had a high TST positivity prevalence compared to their peers in Capricorn (22.0% vs. 7.6%, and 20.5% vs. 2.3%, using TST thresholds of $\geq$5mm and $\geq$10mm respectively). Compared to contacts from Capricorn, household contacts living in Mangaung were more likely to have TST induration $\geq$5mm (odds ratio [OR]: 3.08, 95% credibility interval [CI]: 2.13–4.58) and $\geq$10mm (OR: 4.52, 95% CI: 3.03–6.97). There was a 90% and 92% posterior probability that the odds of TST induration $\geq$5mm (OR: 0.79, 95% CI: 0.56–1.14) and $\geq$10mm (OR: 0.77, 95% CI: 0.53–1.10) respectively were lower in household contacts of HIV-positive compared to HIV-negative index cases.

## Conclusions

High TST induration positivity, especially among young children and people living in Mangaung indicates considerable TB transmission despite high antiretroviral therapy coverage. Household contact of HIV-positive index TB cases were less likely to have evidence of latent TB infection than contacts of HIV-negative index cases.

## Introduction

An estimated one-quarter of the world's population are latently-infected with *Mycobacterium tuberculosis*,[1] and are at risk of progression to active disease.[2,3] To achieve the End-TB strategy, the World Health Organization (WHO) has prioritised the identification and delivery of tuberculosis (TB) preventive interventions to people at high-risk of infection.[4] In sub-Saharan Africa, people living with human immunodeficiency virus (HIV) infection continue to have substantially-higher incidence of active TB disease than people without HIV, despite high levels of population coverage of antiretroviral therapy (ART) in many settings.[5,6] TB preventive therapy significantly reduces the risk of progression to active disease for both HIV-positive and HIV-negative individuals,[7–9] and newer more tolerable and shorter regimens are becoming available through national HIV and TB programmes.[10] However, coverage of TB preventive therapy remains unacceptably low and innovative approaches to prioritise groups at the highest risk of disease are urgently needed.[11]

People exposed to a person with infectious pulmonary TB within the home (household contacts) are likely to share risk factors for TB with index cases, such as poverty, living and environmental conditions and health determinants, including HIV status, nutrition and access to healthcare.[12] Indeed, data from several settings demonstrates high prevalence and incidence of TB disease in household contacts following diagnosis of an index patients with TB.[13–15] The evidence for the effectiveness and cost-effectiveness of household contact screening and treatment in sub-Saharan Africa is limited [16] because of the uncertainty over the role of community transmission outside the household.[17] However, in high-income settings public health approaches to TB contact-tracing have prioritised the most susceptible individuals and those with the greatest cumulative exposure for preventive interventions.[18] In high HIV prevalence settings, there remains considerable uncertainty around which index case-, household contact-, and community-level factors that are most important in determining risk of TB infection.

Using data collected prospectively from participants in the intervention arm of a household cluster-randomised trial being conducted in South Africa, we set out to describe the prevalence

of latent TB infection among household contacts of pulmonary TB index cases, and thereby identify population groups that may receive the greatest benefit and could be targeted in efforts to expand access to TB preventive therapy, or for roll-out of novel TB vaccines. Using a Bayesian multilevel modelling analysis approach, we additionally aimed to provide evidence for how shared household risk factors between index cases (particularly HIV status) and household contacts interact to determine household contact prevalence of latent TB infection within provinces with differing TB and HIV epidemics.

## Materials and methods

### Study setting and participants

We conducted an epidemiological analysis of data collected during a household cluster randomised trial (ISRCTN16006202) conducted in two provinces of South Africa investigating the effect of intensified home screening and linkage to treatment for TB and HIV on TB-free survival.[19] We purposefully selected two study sites with very different annual TB incidence. The study sites were Mangaung Municipality in the Free State (2016 population: 787,803); and Capricorn District in Limpopo Province (2016 population: 1,330,436).[20] Antenatal HIV seroprevalence and the annual TB incidence for 2015 in Mangaung and in Capricorn were 31.7% and 616/100,000, and 21.6% and 328/100,000, respectively.[21]

At both sites index TB cases were identified by visiting hospital wards and clinics, reviewing routine reporting of TB cases in public-sector health facilities (TB registers and notifications), and from lists of specimens that were either smear-positive or culture-positive for *M.tuberculosis* generated by public sector laboratories. Index TB cases of all ages who had been diagnosed with TB within the preceding six-weeks were eligible to participate, providing they were a permanent resident of one of the study districts, and with no plans to relocate during the study. We excluded TB cases older than seven years if they did not have a microbiologically-confirmed diagnosis of pulmonary TB; children seven years-or-younger were eligible if they had either microbiologically-confirmed or microbiologically-unconfirmed TB, diagnosed by a doctor. Severely-ill and deceased patients were included if consent to participate was obtained from a close family member. We excluded TB cases who were incarcerated or receiving long-term in-patient care.

Because of the profusion of backyard dwellings, a household was defined as all rooms under a contiguous roofed area linked by doorways or windows through which air could pass. Household contacts of index TB cases were defined as individuals who slept overnight at least once or shared at least two meals in the same household as the index case in the 14 days prior to the index case's diagnosis of TB.

For this analysis, we only included participants randomly allocated to the intervention arm of the trial. In the intervention arm, households received a home visit within 14 days of recruiting the index TB patient, where a household census was undertaken, and a socio-demographic and health questionnaire was completed for each household contact. Household contacts were screened for symptoms of TB, and sputum was collected from participants able to produce it–irrespective of the presence of symptoms–with Xpert and culture requested to be performed at the local National Health Laboratory Service (NHLS) mycobacterial laboratory, and with subsequent linkage to treatment for positive results. HIV testing and counselling with linkage to care and prevention services was offered to all household contacts who did not have previously confirmed HIV infection or who were not taking ART. Isoniazid preventive therapy (IPT) initiation in the household was offered to HIV-positive individuals and children younger than five years without TB symptoms; we also offered TB preventive treatment to HIV-negative household contacts older than five years. Initially, following National TB Guidelines, home

initiation of IPT was based on the presence of a positive tuberculin skin test (TST) for HIV-infected individuals and children under five years of age. Approximately midway through the study having a positive TST result was removed from Guidelines[22] and we then offered home IPT initiation to all children under five years of age and all HIV-infected individuals irrespective of the TST result. However, we continued until study end to offer home IPT initiation for HIV-seronegative individuals with a positive TST.

## Tuberculin skin testing

Tuberculin skin testing using purified protein derivative (PPD; from a variety of manufacturers, owing to limited global supply) was offered to all household contacts. TST was administered by study nurses who received study-specific training. Quality of TST testing was evaluated by supervisors to ensure standardization at both sites. An intradermal injection of 0.1 milliliter of PPD was placed in the volar aspect of the left forearm to raise a visible bleb. Household contacts were visited 48–72 hours after placement for reading. The presence of induration was recorded, and two dots were marked at each margin on the long axis of the forearm. The induration diameter was then measured using a millimeter ruler.[23]

## Ethical considerations

All index TB cases (or their representative if deceased or severely unwell) and household contacts provided written informed consent to participate in the study. Consent to participate in research was obtained for children <18 years from their parent or guardian, and assent was obtained from children 7–18 years. Approvals were obtained from ethical review committees of the University of Witwatersrand and London School of Hygiene and Tropical Medicine.

## Statistical methods

We summarised the characteristics of index cases and their household contacts using medians, ranges and proportions, and compared the two study sites. To account for missing values in the HIV status of index cases (4.9%) and HIV status of household contacts (5.0%), we did multiple imputation using additive regression, bootstrapping, and predictive mean matching.[24]

To investigate whether index case HIV status was associated with evidence of latent TB infection, we fitted two multilevel Bayesian regression models with Bernoulli likelihood distribution to the TST data. The first model investigated the proportion of household contacts with TST induration ≥5mm, and the second the proportion of household contacts with TST induration ≥10mm. TST thresholds were based on conventionally-used definitions for latent TB infection. We plotted a directed acyclic graph to describe the putative causal pathway between index case HIV-positivity and household contact latent TB infection, and to identify the minimal sufficient adjustment set of covariates for estimating the direct effect of index case HIV status on latent TB infection (S1 Fig). Random intercepts for each household were included to allow partial pooling of estimates and to account for clustering of characteristics within households.[25] We placed weakly informative priors on intercepts (normal[mean = 0, sd = 2]), beta coefficients (normal[mean = 0, sd = 2]), and on the standard deviation of random intercepts (half-Cauchy[0,1]). Parameters and their credible intervals were estimated using Hamiltonian Markov chain Monte Carlo procedures implemented in Stan, interfaced through the 'brms 2.9.0' package in R.[26] The percentage and 95% uncertainty intervals of household contacts with TST induration reactions ≥5mm and ≥10mm were estimated by drawing samples from model posterior distributions, and fitting to index case age distributions, stratified by study site and index case HIV status. Model checking included inspection of prior predictive

distributions, inspection of trace plots, estimation of the number of effective samples per iteration, and estimation of R̂ statistics.[27]

## Data sharing

Code and datasets to reproduce analysis are available to download from https://github.com/petermacp/tstsa

## Results

### Characteristics of index patients and household contacts

A total of 924 index TB patients were included in this analysis. Overall, 60% (559/924) of index patients were male, and the median age was 37 years (interquartile range [IQR]: 28–47 years) (Table 1); 2.9% (27/924) TB patients were included after their death, and 98% (897/924) had microbiologically-confirmed TB. HIV prevalence among index TB patients was 56%, and 65% of HIV-positive TB patients were taking ART, with a median CD4 cell count of (133 cells/μl, IQR: 53–330). Index patients from Mangaung were more likely to be current tobacco smokers (26% vs. 22%), alcohol drinkers (33% vs. 26%), and have unknown HIV status (6.2% vs. 2.5%).

A total of 2,985 household contacts were identified from 924 intervention households, with a median of 4.0 (range: 1.0–14.0) contacts per index patient in both Mangaung and Capricorn. In both sites, 38% of household contacts were male, and the overall median age was 17 years (range: 0–98 years). Household contacts from Mangaung were more likely to share a bedroom with the index TB patient compared to contacts in Capricorn (23% vs. 9%). HIV prevalence was higher in Mangaung compared to in Capricorn (14% vs. 10%), and ART coverage among HIV-positive household contacts was high in both sites (86% and 89% respectively). In Mangaung, 47% of household contacts reported that they spent "most of the day" with the index case, compared to only 33% in Capricorn.

### Tuberculin skin testing

Overall, 2,725/2,985 (91.3%) of household contacts underwent TST. Characteristics of household contacts who did and did not receive TST were similar, but with some minor differences (S1 Table). A greater proportion of household contacts in Capricorn (95.8%) compared to Mangaung (87.8%) received TST testing. Additionally, HIV-positive (84.0%) household contacts were less likely to receive TST compared to HIV-negative household contacts (94.0%). HIV-positive household contacts taking ART (73.4%) were less likely to receive TST compared to those not taking ART (81.4%). Sixty-four percent of household contacts had no induration. Seventeen percent (458/2,725, 16.8%) of household contacts had an induration diameter of ≥5mm; and 13% (359/2725, 13.1%) had an induration diameter of ≥10mm.

At both TST induration diameter cut-offs (≥5mm, ≥10mm), the percentage of household contacts with a positive TST reaction showed an S-shaped distribution across age groups with a higher percentage in the 0–4 year old group compared to in childhood and early adolescence, before increasing rapidly in late adolescence and early adulthood (Fig 1), although the early childhood peak was more apparent in Mangaung than in Capricorn. TST positivity was consistently higher across all contact age groups in the Mangaung site compared to the Capricorn site.

There were substantial differences between study sites in the percentage of TST positive household contacts when examined by key index TB case and household contact characteristics (Table 2). Across all characteristics (excluding among household contacts who were previous smokers, where numbers were small), the percentage of household contacts with either a

**Table 1. Characteristics of index cases (top), dwellings (centre), and household contacts (bottom) by study site.**

| | Mangaung | Capricorn | Total |
|---|---|---|---|
| **Index TB cases** | **(N = 517)** | **(N = 407)** | **(N = 924)** |
| **Sex** | | | |
| Female | 191 (36.9%) | 174 (42.8%) | 365 (39.5%) |
| Male | 326 (63.1%) | 233 (57.2%) | 559 (60.5%) |
| **Age (years, median, IQR)** | 37 (28, 48) | 37 (29, 47) | 37 (28, 47) |
| **Vital status at recruitment** | | | |
| Alive | 504 (97.5%) | 393 (96.6%) | 897 (97.1%) |
| Deceased | 13 (2.5%) | 14 (3.4%) | 27 (2.9%) |
| **Microbiological TB status** | | | |
| Not microbiologically-confirmed TB | 8 (1.5%) | 14 (3.4%) | 22 (2.4%) |
| Microbiologically-confirmed TB | 509 (98.5%) | 393 (96.6%) | 902 (97.6%) |
| **Tobacco smoking** | | | |
| Never smoked | 253 (48.9%) | 267 (65.6%) | 520 (56.3%) |
| Previous smoker | 127 (24.6%) | 49 (12.0%) | 176 (19.0%) |
| Current smoker | 137 (26.5%) | 91 (22.4%) | 228 (24.7%) |
| **HIV status** | | | |
| HIV-negative | 207 (40.0%) | 178 (43.7%) | 385 (41.7%) |
| HIV-positive | 278 (53.8%) | 219 (53.8%) | 497 (53.8%) |
| HIV-unknown | 32 (6.2%) | 10 (2.5%) | 42 (4.5%) |
| **Duration of cough (days, median, IQR)** | 30 (10, 60) | 30 (0, 60) | 30 (7, 60) |
| **Dwellings** | **(N = 517)** | **(N = 407)** | **(N = 924)** |
| **Dwelling type** | | | |
| Brick/concrete house, apartment | 443 (85.7%) | 402 (98.8%) | 845 (91.5%) |
| Traditional/shack/garage/1room | 74 (14.3%) | 5 (1.2%) | 79 (8.5%) |
| **Rooms in dwelling (median, range)** | 4.0 (1.0, 10.0) | 6.0 (1.0, 22.0) | 5.0 (1.0, 22.0) |
| **Windows in dwelling (median, range)** | 4.0 (0.0, 15.0) | 8.0 (0.0, 27.0) | 5.0 (0.0, 27.0) |
| **Tobacco smokers in household (median, range)** | 0.0 (0.0, 4.0) | 0.0 (0.0, 6.0) | 0.0 (0.0, 6.0) |
| **Household contacts** | **(N = 1687)** | **(N = 1298)** | **(N = 2985)** |
| **Contacts per household (median, range)** | 4.0 (1.0, 14.0) | 4.0 (1.0, 14.0) | 4.0 (1.0, 14.0) |
| **Age group of household contact** | | | |
| 0–4 years | 196 (15.1%) | 249 (14.8%) | 445 (14.9%) |
| 5–9 years | 206 (15.9%) | 274 (16.2%) | 480 (16.1%) |
| 10–15 years | 174 (13.4%) | 267 (15.8%) | 441 (14.8%) |
| 15–24 years | 219 (16.9%) | 267 (15.8%) | 486 (16.3%) |
| 25–34 years | 139 (10.7% | 176 (10.4%) | 315 (10.6%) |
| 35–44 years | 78 (6.0%) | 133 (7.9%) | 211 (7.1%) |
| 45–54 years | 84 (6.5%) | 106 (6.3%) | 190 (6.4%) |
| 55+ years | 202 (15.6%) | 215 (12.7%) | 417 (14.0%) |
| **Household contact sex** | | | |
| Female | 1045 (61.9%) | 803 (61.9%) | 1848 (61.9%) |
| Male | 642 (38.1%) | 495 (38.1%) | 1137 (38.1%) |
| **Household contact age (median, range)** | 16.0 (0.0, 95.0) | 17.0 (0.0, 98.0) | 17.0 (0.0, 98.0) |
| **Time spent with index case** | | | |
| Missing | 1 | 2 | 3 |
| Every now and again | 123 (7.3%) | 241 (18.6%) | 364 (12.2%) |
| Part of the day | 773 (45.8%) | 628 (48.5%) | 1401 (47.0%) |
| Most of the day | 790 (46.9%) | 427 (32.9%) | 1217 (40.8%) |

*(Continued)*

**Table 1.** (Continued)

|  | Mangaung | Capricorn | Total |
|---|---|---|---|
| **Index TB cases** | **(N = 517)** | **(N = 407)** | **(N = 924)** |
| **Shared bedroom with index case** |  |  |  |
| No | 1303 (77.2%) | 1184 (91.2%) | 2487 (83.3%) |
| Yes | 384 (22.8%) | 114 (8.8%) | 498 (16.7%) |
| **Tobacco smoking** |  |  |  |
| Never smoked | 1498 (88.8%) | 1200 (92.4%) | 2698 (90.4%) |
| Current smoker | 159 (9.4%) | 83 (6.4%) | 242 (8.1%) |
| Previous smoker | 30 (1.8%) | 15 (1.2%) | 45 (1.5%) |
| **HIV status of household contact** |  |  |  |
| Missing | 0 | 3 | 3 |
| HIV negative | 1349 (80.0%) | 1136 (87.7%) | 2485 (83.3%) |
| HIV positive | 223 (13.2%) | 126 (9.7%) | 349 (11.7%) |
| HIV unknown | 115 (6.8%) | 33 (2.5%) | 148 (5.0%) |

TST induration ≥5mm or ≥10mm were higher in Mangaung. In Capricorn 11.4% of household contacts of HIV-positive index cases and 9.8% of household contacts of HIV-negative index cases had TST induration of ≥5mm; however, in Mangaung 19.7% of household contacts of HIV-positive index cases and 24.8% of household contacts of HIV-negative index cases had TST duration of ≥5mm. This difference in percentage by site was apparent at the ≥10mm threshold. Whereas in Capricorn TST positivity rates were similar for male (10.9%) and female (10.5%) household contacts, in Mangaung female (23.4%) contacts were more likely to be TST positive than male contacts (19.6%) at the ≥5mm thresholds. Household contacts who were current smokers had a higher percentage of TST positivity at both the ≥5mm and ≥10mm thresholds in both Capricorn and Mangaung.

## Associations with TST positivity

In the directed acyclic graph (S1 Fig), study site and index case HIV status were identified as the minimal sufficient adjustment set of covariates for estimating the direct effect of index case HIV status on latent TB infection. Models fitted to data (Fig 2) showed marked differences between study sites in the proportion of household contacts with TST induration ≥5mm and ≥10mm, with contacts from Mangaung having consistently higher percentages of positivity compared to contacts from Mangaung. Both study sites showed that the probability of household contact TST positivity was highest where the index TB case was young and decreased with index case, but these trends were most pronounced among contacts from Mangaung. Across all index case ages, the mean proportion of contacts with a TST reaction ≥5mm and ≥10mm were consistently higher where the index case was HIV-negative.

In multivariable regression analysis, there was strong evidence that a greater percentage of household contacts resident in Mangaung compared to Capricorn had TST ≥5mm (odds ratio [OR]: 3.08, 95% credible interval [CI]: 2.13–4.58) and ≥10mm (OR: 4.52, 95% CI: 3.03–6.97). Where the index TB case was HIV positive, household contacts were less likely to be TST positive at the ≥5mm threshold (OR: 0.79, 95% CI: 0.56–1.14), or at the ≥10mm thresholds (0.77, 95% CI: 0.53–1.10). The posterior probability that the odds of TST induration ≥5mm were lower in household contacts of HIV-positive compared to HIV-negative index cases was 90%; for TST induration ≥10mm it was 92%.

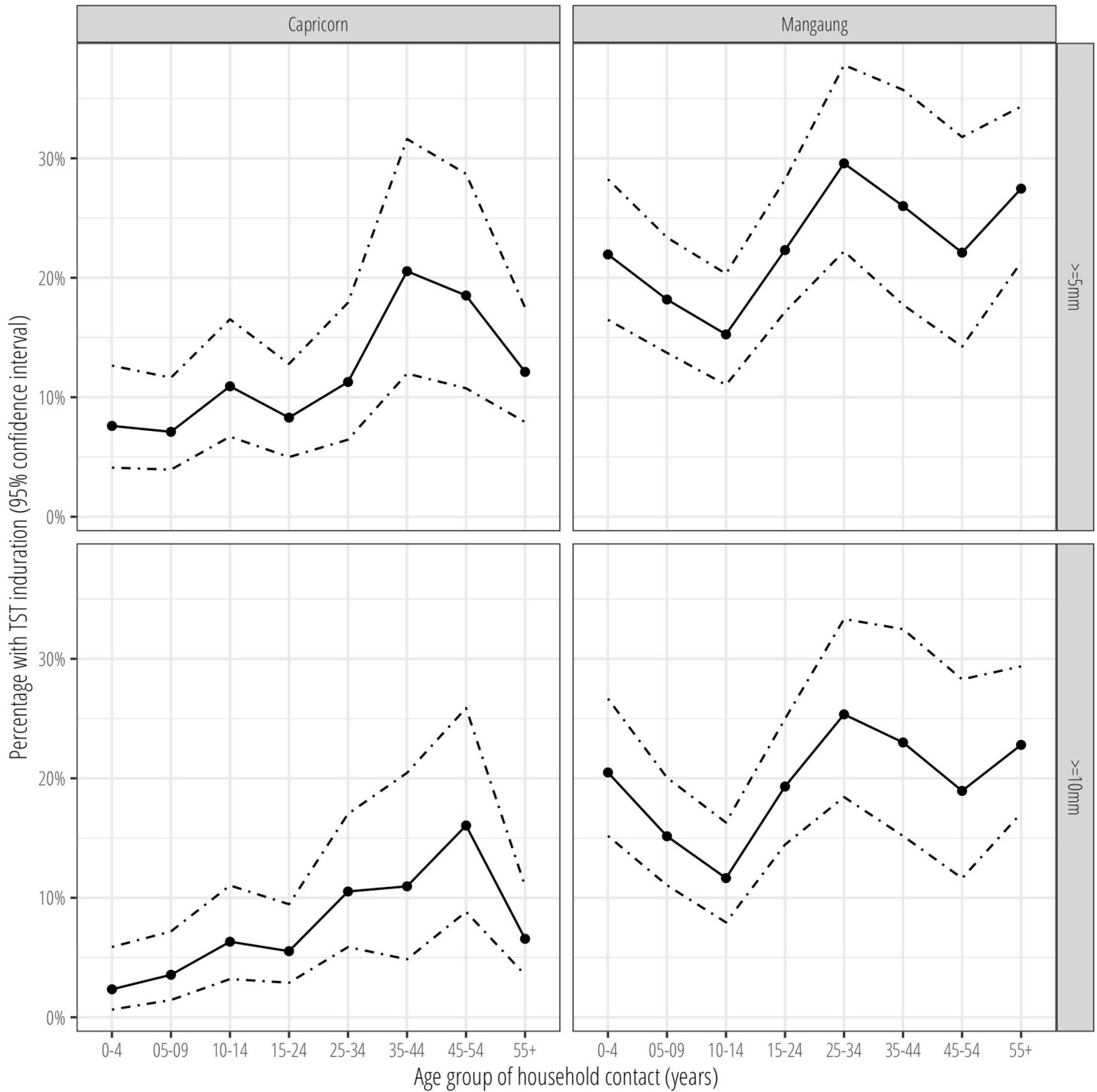

**Fig 1. Tuberculin skin test positivity (≥5mm, ≥10mm) by household contact age group in Capricorn and Mangaung, South Africa.** 95% confidence intervals estimated using the binomial exact method.

Each year increase in the age of the index case was associated with a 2% decrease in the odds of TST positivity at the ≥5mm thresholds (OR: 0.98, 95% CI: 0.96–0.99), and a 2% decrease at the ≥10mm threshold (OR: 0.98, 95% CI: 0.97–0.99).

**Table 2. Tuberculin skin test positivity by index TB case and household contact characteristics.**

| Characteristic | TST threshold | Capricorn n/N, (%, 95% CI)* | Mangaung n/N, (%, 95% CI)* |
|---|---|---|---|
| Index TB case female | ≥5mm | 64/540 (11.9%, 9.2%-14.9%) | 131/599 (21.9%, 18.6%-25.4%) |
| Index TB case male | ≥5mm | 69/704 (9.8%, 7.7%-12.2%) | 194/882 (22.0%, 19.3%-24.9%) |
| Index TB case HIV-negative | ≥5mm | 54/553 (9.8%, 7.4%-12.5%) | 161/648 (24.8%, 21.6%-28.4%) |
| Index TB case HIV-positive | ≥5mm | 79/691 (11.4%, 9.2%-14.0%) | 164/833 (19.7%, 17.0%-22.6%) |
| Index TB case not micro-confirmed TB | ≥5mm | 3/51 (5.9%, 1.2%-16.2%) | 7/27 (25.9%, 11.1%-46.3%) |
| Index TB case micro-confirmed TB | ≥5mm | 130/1193 (10.9%, 9.2%-12.8%) | 318/1454 (21.9%, 19.8%-24.1%) |
| Contact female | ≥5mm | 81/769 (10.5%, 8.5%-12.9%) | 216/925 (23.4%, 20.7%-26.2%) |
| Contact male | ≥5mm | 52/475 (10.9%, 8.3%-14.1%) | 109/556 (19.6%, 16.4%-23.2%) |
| Contact HIV-negative | ≥5mm | 113/1129 (10.0%, 8.3%-11.9%) | 274/1286 (21.3%, 19.1%-23.6%) |
| Contact HIV-positive | ≥5mm | 20/115 (17.4%, 11.0%-25.6%) | 51/195 (26.2%, 20.1%-32.9%) |
| Contact with index case most of day | ≥5mm | 43/406 (10.6%, 7.8%-14.0%) | 142/663 (21.4%, 18.4%-24.7%) |
| Contact with index case part of day | ≥5mm | 65/609 (10.7%, 8.3%-13.4%) | 154/701 (22.0%, 19.0%-25.2%) |
| Contact with index case rarely | ≥5mm | 25/229 (10.9%, 7.2%-15.7%) | 29/117 (24.8%, 17.3%-33.6%) |
| Shares bedroom with index case | ≥5mm | 112/1134 (9.9%, 8.2%-11.8%) | 244/1162 (21.0%, 18.7%-23.5%) |
| Doesn't share bedroom with index case | ≥5mm | 21/110 (19.1%, 12.2%-27.7%) | 81/319 (25.4%, 20.7%-30.5%) |
| Contact never smoked | ≥5mm | 115/1147 (10.0%, 8.3%-11.9%) | 287/1318 (21.8%, 19.6%-24.1%) |
| Contact current smoker | ≥5mm | 15/82 (18.3%, 10.6%-28.4%) | 33/134 (24.6%, 17.6%-32.8%) |
| Contact previous smoker | ≥5mm | 3/15 (20.0%, 4.3%-48.1%) | 5/29 (17.2%, 5.8%-35.8%) |
| Index TB case female | ≥10mm | 41/540 (7.6%, 5.5%-10.2%) | 109/599 (18.2%, 15.2%-21.5%) |
| Index TB case male | ≥10mm | 41/704 (5.8%, 4.2%- 7.8%) | 168/882 (19.0%, 16.5%-21.8%) |
| Index TB case HIV-negative | ≥10mm | 31/553 (5.6%, 3.8%- 7.9%) | 141/648 (21.8%, 18.6%-25.1%) |
| Index TB case HIV-positive | ≥10mm | 51/691 (7.4%, 5.5%- 9.6%) | 136/833 (16.3%, 13.9%-19.0%) |
| Index TB case not micro-confirmed TB | ≥10mm | 2/51 (3.9%, 0.5%-13.5%) | 6/27 (22.2%, 8.6%-42.3%) |
| Index TB case micro-confirmed TB | ≥10mm | 80/1193 (6.7%, 5.4%- 8.3%) | 271/1454 (18.6%, 16.7%-20.7%) |
| Contact female | ≥10mm | 51/769 (6.6%, 5.0%- 8.6%) | 181/925 (19.6%, 17.1%-22.3%) |
| Contact male | ≥10mm | 31/475 (6.5%, 4.5%- 9.1%) | 96/556 (17.3%, 14.2%-20.7%) |
| Contact HIV-negative | ≥10mm | 67/1129 (5.9%, 4.6%- 7.5%) | 232/1286 (18.0%, 16.0%-20.3%) |
| Contact HIV-positive | ≥10mm | 15/115 (13.0%, 7.5%-20.6%) | 45/195 (23.1%, 17.4%-29.6%) |
| Contact with index case most of day | ≥10mm | 24/406 (5.9%, 3.8%- 8.7%) | 122/663 (18.4%, 15.5%-21.6%) |
| Contact with index case part of day | ≥10mm | 40/609 (6.6%, 4.7%- 8.8%) | 133/701 (19.0%, 16.1%-22.1%) |
| Contact with index case rarely | ≥10mm | 18/229 (7.9%, 4.7%-12.1%) | 22/117 (18.8%, 12.2%-27.1%) |
| Shares bedroom with index case | ≥10mm | 72/1134 (6.3%, 5.0%- 7.9%) | 205/1162 (17.6%, 15.5%-20.0%) |
| Doesn't share bedroom with index case | ≥10mm | 10/110 (9.1%, 4.4%-16.1%) | 72/319 (22.6%, 18.1%-27.6%) |
| Contact never smoked | ≥10mm | 72/1147 (6.3%, 4.9%- 7.8%) | 244/1318 (18.5%, 16.5%-20.7%) |
| Contact current smoker | ≥10mm | 9/82 (11.0%, 5.1%-19.8%) | 28/134 (20.9%, 14.4%-28.8%) |
| Contact previous smoker | ≥10mm | 1/15 (6.7%, 0.2%-31.9%) | 5/29 (17.2%, 5.8%-35.8%) |

*95% confidence interval estimated using binomial exact method

## Discussion

In this analysis of >2,500 household contacts of index TB patients in the intervention arm of a cluster-randomised trial in two provinces of South Africa, we found a high prevalence of latent TB infection, with contacts living in Mangaung (a locality with very high annual TB burden), older contacts, and contacts of microbiologically-confirmed index TB cases having a greater prevalence of infection. In contrast, we found that household contacts of HIV-positive index TB cases were less likely to have evidence of latent TB infection compared to contacts of HIV-negative index TB cases.

## A) Percentage (95% credible interval) of household contacts with TST induration ≥ 5mm

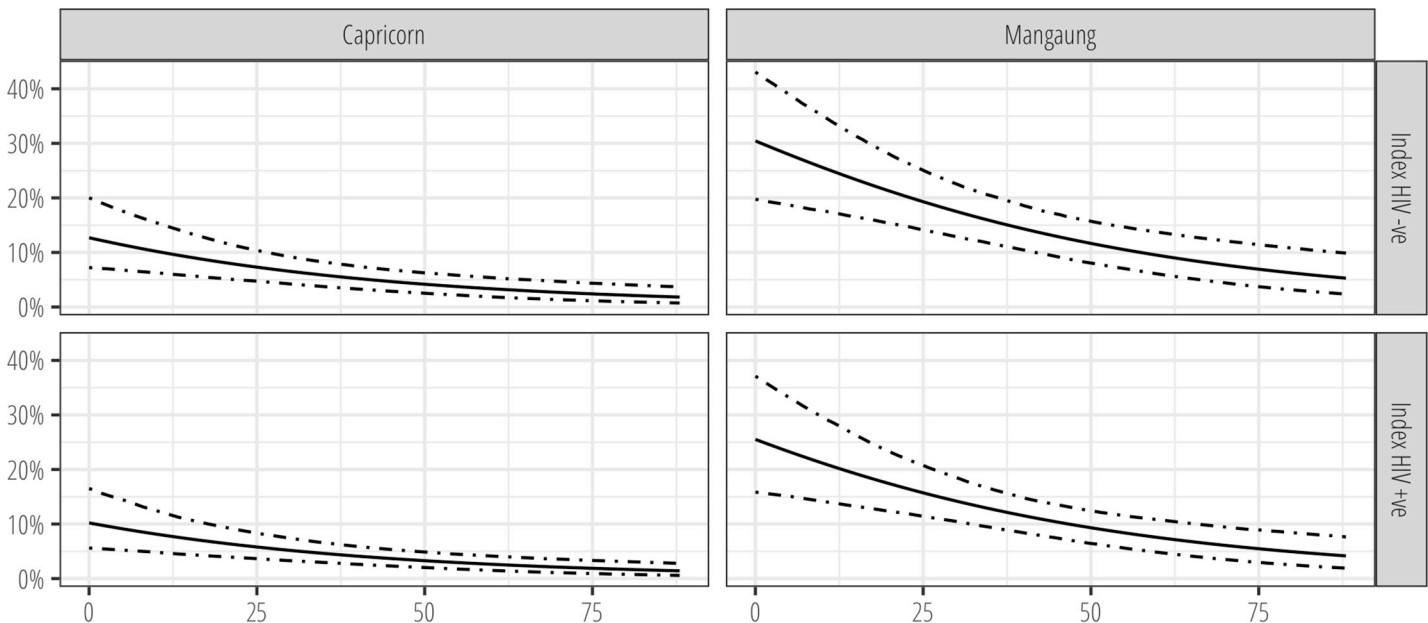

## B) Percentage (95% credible interval) of household contacts with TST induration ≥ 10mm

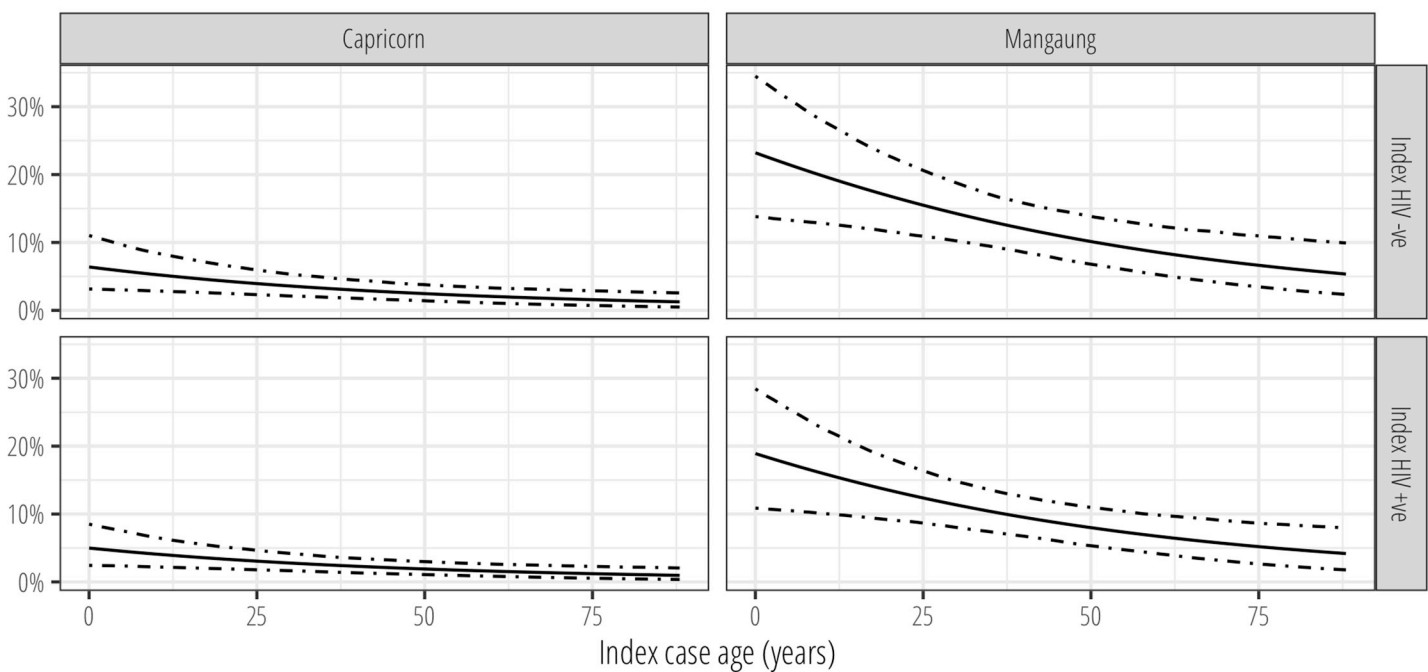

**Fig 2. Model-fitted percentage of household contacts with tuberculin skin test induration ≥5mm and ≥10mm by index TB case age and study site.** Estimated from fitted predictions from hierarchical multivariable Bernoulli regression model, with random intercepts for households.

Of particular concern was the extremely high prevalence of latent TB infection among children aged 0–4 years in Mangaung, with 22% and 20% having indurations of ≥5mm and

≥10mm respectively. Similar rates were reported in a township in the Western Cape [28] and in Matlosana in younger individuals.[29] Evidence increasingly shows that young children (<5 years old) are predominantly exposed to infectious TB outside their immediate household.[17] A high prevalence of latent TB infection among this group is indicative of intense community transmission and high prevalence of undiagnosed infectious TB,[30] meaning that urgent public health action is required to increase case detection and protect vulnerable groups and individuals. We are evaluating an intervention to protect household contacts and improve TB free survival in contacts of TB index patients in the parent cluster randomised trial; results will be reported once the trial has concluded.

We found that, after adjusting for important confounders, household contacts of HIV-positive index cases had substantially lower odds of latent TB infection at both the ≥5mm and ≥10mm thresholds, with differences particularly apparent when compared between study sites. In the Capricorn study site–which included many participants living in rural areas, has a lower HIV prevalence, and almost half the TB annual TB incidence of Mangaung[21]–the proportion of household contacts with immunological evidence of latent TB infection was lower than in Mangaung (which is characterised by large, densely-populated urban and peri-urban townships and high HIV prevalence) across all age bands. In both sites, the prevalence of TST positivity began to decline after approximately age 34–44 years, and may represent a survivorship effect with disproportionate population mortality among younger adults with TB infection over the past 10–20 years who have not survived into older age due to HIV-related mortality.[31]

There is uncertainty over the role of HIV in the infectiousness of pulmonary tuberculosis, particularly with immune reconstitution due to ART.[32] This is partly because of the lack of a robust test for diagnosis of recent TB infection in settings where community force of infection is high, and data suggesting that HIV infection may lead to a shorter duration of symptomatic pulmonary disease and with less cavitation and lower bacillary load in people living with HIV. [33] In a high-TB burden community, TST induration in a household contact may represent infection as a consequence of household exposure from an index case, infection from other community contacts, or both. In this study, we found that contacts of HIV-positive index cases had a lower prevalence of TST positivity compared to contacts of HIV-negative household contacts, even when stratified by study sites with markedly different epidemiological patterns of HIV and TB. Although by no means conclusive and assuming that contacts of HIV-positive and HIV-negative index cases had similar patterns of exposure outside of the household, this suggests that HIV-positive TB patients with pulmonary tuberculosis are less infectious than HIV-negative TB patients. A greater understanding of the possible reasons for this is required, including through comparison of the rate of generation of infectious quanta between HIV-positive and HIV-negative people, and although challenging, prospective cohort studies of incidence in latent TB infection (perhaps using newer, more accurate tests) combined with epidemiological and social mixing data should be considered.

From a public health perspective, a number of recommendations arise from this analysis. In high HIV prevalence settings, substantially greater evidence for the effectiveness and cost-effectiveness of household contact screening vs. current standard of care (primarily delivered in conjunction with passive detection of active TB cases, with few resources available for contact tracing activities, and occasionally through active case finding activities) is needed. Shortened, more tolerable TB preventive treatment regimens may significantly alter what is feasible and acceptable to deliver to individuals and communities,[10] and operational research to develop, refine and evaluate new models will be critical to achieving success. Importantly, as data from the current analysis demonstrates, a single delivery approach generalised across all population groups in highly-differentiated epidemiological settings are unlikely to be

successful. Instead, local public health authorities need local-relevant data on the trends and patterns of infection and disease within their populations to inform the design of a suite of TB prevention interventions. This will require a renewal of local public health capacity, as well as strengthened links within and between communities.

There are a number of limitations to this analysis. There were differences in characteristics between individuals who received and did not receive TST in the study, with HIV-positive individuals less likely to complete testing. This could potentially have resulted in selection bias, although it is uncertain in which direction this could have influenced estimates. Although ART status of household contacts was recorded, we were unable to evaluate the interaction between HIV status, ART and age on TST induration diameter due to the small numbers of young children with HIV in this epidemiologically-important age group. PPD was intermittently available during the trial period due to a global shortage, and we had to purchase PPD from different suppliers. We didn't record Bacille Calmette–Guérin (BCG) vaccination status of household contacts, although coverage of BCG vaccination is high in South Africa. Although standardised training and regular quality assessment of TST procedures was provided (including periods where study coordinators swapped between sites), it is possible that performance of TST (which is subject to operator and measurement error) may have varied between sites. We evaluated the effect of intensity of exposure to index cases by categorising into self-reported groups; although very resource-intensive, future studies may wish to measure exposure prospectively using diaries or contact sensor approaches. Our estimate of the effect of index case HIV-positive status on TST positivity in household contacts can only be interpreted as causal under the assumption of a correctly specified DAG model. Moreover, unmeasured confounding may be an issue, particularly due to latent variables identified but uncontrolled for in the DAG. Finally, in the absence of a gold-standard test for latent TB infection and the limitations of TST, our estimates of prevalence are likely to be subject to some misclassification bias.

In conclusion, we found strong evidence of a very high burden of latent TB infection among household contacts of TB cases in two provinces of South Africa, with worryingly high infection rates among young children in Mangaung, Free State. We found evidence that household contacts of HIV-positive index cases were less likely to have latent TB infection than contacts of HIV-negative index cases. In addition to urgent public health action to reduce TB transmission, new approaches are required to reach population groups at greatest risk and deliver effective TB preventive treatment.

## Supporting information

**S1 Table. Characteristics of household contacts who did and did not receive tuberculin skin testing.** Putative causal pathways are shown by arrowed lines. Unmeasured variables are filled in white. Using the backdoor criteria, study site, household contact sex and HIV status, and index case age, sex, HIV status and microbiological TB status were identified as the minimal sufficient adjustment set of covariates for estimating the direct effect of index case HIV status on latent TB infection.
(DOCX)

**S1 Fig. Directed acyclic graph to describe the putative causal relationship between index case HIV status and household contact latent TB infection.**
(TIFF)

## Author Contributions

**Conceptualization:** Peter MacPherson, Andrew Ratsela, Sanjay Lala, Ebrahim Variava, Jonathan E. Golub, Emily L. Webb, Neil A. Martinson.

**Formal analysis:** Peter MacPherson, Emily L. Webb.

**Funding acquisition:** Peter MacPherson, Andrew Ratsela, Ebrahim Variava, Emily L. Webb, Neil A. Martinson.

**Investigation:** Limakatso Lebina, Kegaugetswe Motsomi, Zama Bosch, Minja Milovanovic.

**Project administration:** Limakatso Lebina, Kegaugetswe Motsomi, Zama Bosch, Minja Milovanovic.

**Writing – original draft:** Peter MacPherson, Minja Milovanovic, Emily L. Webb, Neil A. Martinson.

**Writing – review & editing:** Peter MacPherson, Limakatso Lebina, Kegaugetswe Motsomi, Zama Bosch, Minja Milovanovic, Andrew Ratsela, Sanjay Lala, Ebrahim Variava, Jonathan E. Golub, Emily L. Webb, Neil A. Martinson.

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
