## [Decision Letter · Decision Letter 0]

17 Dec 2019

PONE-D-19-29656

Prevalence and risk factors for latent tuberculosis infection among household contacts of index cases in two South African provinces: analysis of baseline data from a cluster-randomised trial

PLOS ONE

Dear Dr. MacPherson,

Thank you for submitting your manuscript to PLOS ONE. After careful consideration, we feel that it has merit but does not fully meet PLOS ONE’s publication criteria as it currently stands. Therefore, we invite you to submit a revised version of the manuscript that addresses the points raised during the review process.

The two reviewers have made some helpful comments and suggestions on the manuscript - I would encourage you to address these in full. In particular, I do agree with the point from reviewer #1 that there could perhaps be stronger justification for analyzing TST induration as a continuous variable in your model.

In terms of the first major comment from reviewer #2, I'm not quite sure if they are just suggesting some more discussion of the likely direction of bias based on the minor differences between those with vs without TST results, or suggesting imputing outcome data for the missing TST results. I would suggest the former would be fine, but I don't think the latter is necessary.   

We would appreciate receiving your revised manuscript by Jan 31 2020 11:59PM. To enhance the reproducibility of your results, we recommend that if applicable you deposit your laboratory protocols in protocols.io, where a protocol can be assigned its own identifier (DOI) such that it can be cited independently in the future. For instructions see: http://journals.plos.org/plosone/s/submission-guidelines#loc-laboratory-protocols

We look forward to receiving your revised manuscript.

Kind regards,

Richard John Lessells, BSc, MBChB, MRCP, DTM&H, DipHIVMed, PhD

Academic Editor

PLOS ONE

Journal Requirements:

Reviewers' comments:

Reviewer's Responses to Questions

**Comments to the Author**

1. Is the manuscript technically sound, and do the data support the conclusions?

Reviewer #1: Yes

Reviewer #2: Yes

2. Has the statistical analysis been performed appropriately and rigorously? 

Reviewer #1: No

Reviewer #2: Yes

3. Have the authors made all data underlying the findings in their manuscript fully available?

Reviewer #1: Yes

Reviewer #2: Yes

4. Is the manuscript presented in an intelligible fashion and written in standard English?

Reviewer #1: Yes

Reviewer #2: Yes

5. Review Comments to the Author

Reviewer #1: This a an interesting analysis of a single arm of a clustered randomised clinical trial conducted in two study sites with different annual incidence of TB (Mangaung Municipality in the Free State and Capricorn District in Limpopo Province).

The analysis is well conducted and manuscript easy to read.

I have a number of points that need to be addressed:

1. I am unclear why the length of induration as a continuous variable was used as the primary endpoint. The diameters of 5 and 10 mm have been mentioned as standard clinical cut-off.

Thus, for example what is the clinical relevance of a difference in risk of 3.68 (95% 3.04-4.41) comparing the two sites? Is this a risk or, being the endpoint continuous, a difference in mm? I would have been more relevant to show that the proportion of people with a diameter >5 mm (or >100 mm) was higher in one site compared to the other. All other detected differences are even smaller than 3.7 (mm?). Figure 3 clearly shows means in mm.

2. In the Discussion it is stated that the induration was greater in HIV-positive compared to HIV-negative contacts after controlling for confounding factors. My understanding is that potential confounding was controlled by fitting random intercepts for each household to allow partial pooling of estimates and to account for clustering of characteristics within households. Correct? Going back for example to the comparison by sites, there were clear differences in terms of prevalence of HIV, frequency of contact with the index case as well as modality of sleeping (in the same bed as index case or not). All these are causes of diameter induration but consequences of residency so are more mediators than confounders. How does the multilevel Bayesian zero-inflated Poisson mixture model account for mediation?

3. Lines 236- 249. Four factors are mentioned as associated with diameter induration from fitting a multivariable model in this section: study site, HIV-status of contacts, HIV-status of index case and proportion of the day spent with index case. It is unclear whether these were ‘independent’ associations. Following from point #2 it makes sense to control for study site when evaluating the effect of HIV-positive status (as study site is a confounder). Viceversa, it seems inappropriate to control for HIV-status in the model focusing on study site as the exposure as HIV-status is a consequence of residency and not a cause. Authors should consider fitting different models according to the exposure of interest and control only for confounding factors, not mediator or possible colliders. Finally, what about age? There is not mention of age (strongly associated in univariable analysis) in the multivariable model (Figure 2).

4. There is no discussion of the fact that proportion of day spent with index case showed a significant association with diameter induration but sleeping in the same bed with the index case was not. What are the possible speculations for these findings?

5. Identifying predictors of larger diameter seems the main aim of the analysis. Nevertheless, message appear to be contradictory. Lines 241-243 implies that HIV-status of index case is indeed a predictor. In contrast in the Discussion it is stated that ‘In this study, we found that contacts of HIV-positive index cases had similar median posterior TST induration diameters to contacts of HIV-negative household contacts, even when stratified by study sites with markedly different epidemiological patterns of HIV and TB’. In general, there is a lack of standardisation in the presentation of results (HIV-status and age seem to be the only important predictor from reading the Conclusions of the abstract; HIV-status and duration of contacts with index case in the last paragraph of the Discussion). If the aim of the paper is direct public health efforts to specific target populations the identification of this population is rather confusing

6. Title of Table 1 should be. Characteristics of index cases (top) and household contacts (bottom) by Study site

7. Lines 224-227. From the introducing text a reader would expect to see the comparison of percentage of positivity in the 0-4 years old vs. >4 years old stratified by site, not the difference in proportion of positivity in the 0-4 years old by site alone.

8. Line 255. What was the p-value for this 3-way interaction? Wonder whether instead of describing the effect of age by HIV-status/study would have been more informative to give the differences according to HIV-status by age/study strata?

9. Abstract line 60. Data were collected prospectively as part of a household cluster-randomised trial in two South Africa provinces (Mangaung, Free State, and Capricorn, Limpopo). In intervention group households, TB contacts underwent HIV testing and tuberculin skin testing (TST).

I would rephrase making it clear from the start that this analysis only include the data from the intervention arm. Essentially is a cohort study using a single arm of a trial. This only becomes clear from reading the full paper.

10. Page 16, four lines form the end. Typo mortality instead of morality

11. Page 19. what does BCG stand for? Spell out

12. Figure 1. Add a footnote to clarify that top of the figure is for cut off 5 mm and bottom for 10 mm?

13. Figure 2. Are these mean differences in diameter or RR? RR of what?

14. Figure 3 How can age of house contact be -1? Why is only the range [-1; +3] shown?

Reviewer #2: I commend the authors for the important study on the major public health problem in South Africa. Although study results are compelling, there are few methodological issues which need to be addressed/further explained in order to improve this manuscript.

Major Issue

• Page 18, paragraph 5, limitations: Authors have acknowledged that there were differences between individuals who received and did not TST in the study and these differences could have resulted in selection bias. I would suggest they use the data on those who did not receive TST to investigate how it could have influenced the results.

Minor Issue

• Page 6 line 151: Authors need to describe cut-offs of skin induration for LTBI under Tuberculin skin testing sub-section of Material and Methods section and which guidelines the cut-offs are based on.

In addition, the manuscript is clearly written in a professional manner. Conclusions are presented in an appropriate fashion and are supported by the data If there is a weakness, it is minor methodological issues (as I have noted above).

6. PLOS authors have the option to publish the peer review history of their article (what does this mean?). If published, this will include your full peer review and any attached files.

Reviewer #1: No

Reviewer #2: Yes: Jabulani Ronnie Ncayiyana

---

## [Author Response · Author response to Decision Letter 0]

29 Jan 2020

Wednesday, 29th January 2020

Dear Prof Lessells,

Re: PONE-D-19-29656. Prevalence and risk factors for latent tuberculosis infection among household contacts of index cases in two South African provinces: analysis of baseline data from a cluster-randomised trial

I write on behalf of my co-authors to thank the reviewers for their careful review, and appreciate the opportunity to respond with a revised submission. Please see a point-by-point response below.

We are particularly grateful for the Reviewers’ suggestions for additional analyses and interpretation; these have undoubtedly strengthened the manuscript. In particular, we have: formulated a more specific hypothesis to be tested in the analysis; constructed a directed acyclic graph to specify the putative causal pathway between index TB case HIV status and household contact latent TB infection, accounting for mediating factors and confounders to identify the minimal sufficient adjustment set of covariates for estimating the direct causal effect; and refined statistical modelling approached to estimate posterior distributions against clinically meaningful outcomes (the proportion of household contacts with tuberculin skin test induration ≥5mm and ≥5mm). Together, we hope that these revised analyses are now methodologically rigorous and will provide strong evidence of clinical and public health relevance.

We look forward to your response.

Yours Sincerely,

Peter MacPherson

(On behalf of the authors)

RESPONSE TO REVIEWER 1’S COMMENTS

1. This an interesting analysis of a single arm of a clustered randomised clinical trial conducted in two study sites with different annual incidence of TB (Mangaung Municipality in the Free State and Capricorn District in Limpopo Province). The analysis is well conducted and manuscript easy to read.

We thank the Reviewer for their detailed comments, which are extremely helpful, and have undoubtedly helped strengthen the manuscript.

2. I am unclear why the length of induration as a continuous variable was used as the primary endpoint. The diameters of 5 and 10 mm have been mentioned as standard clinical cut-off.

Thus, for example what is the clinical relevance of a difference in risk of 3.68 (95% 3.04-4.41) comparing the two sites? Is this a risk or, being the endpoint continuous, a difference in mm? I would have been more relevant to show that the proportion of people with a diameter >5 mm (or >100 mm) was higher in one site compared to the other. All other detected differences are even smaller than 3.7 (mm?). Figure 3 clearly shows means in mm.

Thank you for this comment. Having carefully reviewed the Reviewer’s extremely helpful suggestions here and below, we have revised our analysis strategy to model the proportion of household contacts with TST induration ≥5mm and ≥10mm. We had previously attempted to model TST induration as a continuous mixture distribution, but appreciate that this approach may have been difficult to interpret and have had less immediate public health relevance. We believe that our revised analyse provides stronger evidence of greater clinical and public health relevance (see Statistical Methods section – line 188-288 for revised description of analytic approach.).

3. In the Discussion it is stated that the induration was greater in HIV-positive compared to HIV-negative contacts after controlling for confounding factors. My understanding is that potential confounding was controlled by fitting random intercepts for each household to allow partial pooling of estimates and to account for clustering of characteristics within households. Correct? Going back for example to the comparison by sites, there were clear differences in terms of prevalence of HIV, frequency of contact with the index case as well as modality of sleeping (in the same bed as index case or not). All these are causes of diameter induration but consequences of residency so are more mediators than confounders. How does the multilevel Bayesian zero-inflated Poisson mixture model account for mediation? 

Again, we thank the Reviewer for this insightful comment. The Reviewer rightly notes that that some variable previously identified as confounders may have in fact been mediators. To address this comment, before conducting our revised statistical analysis, we undertook two actions. Firstly, we formulated a more-specific testable hypothesis to be addressed by the analysis, namely: “are household contacts of HIV-positive index TB patients less likely to have latent TB infection?”. Secondly, we constructed a directed acyclic graph based on our knowledge and literature of the epidemiology of TB transmission to specify the putative causal pathway between index TB case HIV status and household contact latent TB infection, accounting for mediating factors and confounders and to identify the minimal sufficient adjustment set of covariates for estimating the direct causal effect. This graph can now be seen in Supplemental Figure 1. By explicitly describing the proposed causal pathway between index case HIV status and household contact latent TB infection, we hope that this will have resulted in a more rational approach to variable inclusion in models. Indeed, variables that the Reviewer identified as potential mediators in the relationship were no longer required to be adjusted for in the fully specified model.

To answer the question about partial pooling – yes, including a random intercept term for households allows information to be shared between households, reducing uncertainty whilst accounting for the clustering of characteristics between households.

4. Lines 236- 249. Four factors are mentioned as associated with diameter induration from fitting a multivariable model in this section: study site, HIV-status of contacts, HIV-status of index case and proportion of the day spent with index case. It is unclear whether these were ‘independent’ associations. Following from point #2 it makes sense to control for study site when evaluating the effect of HIV-positive status (as study site is a confounder). Viceversa, it seems inappropriate to control for HIV-status in the model focusing on study site as the exposure as HIV-status is a consequence of residency and not a cause. Authors should consider fitting different models according to the exposure of interest and control only for confounding factors, not mediator or possible colliders. Finally, what about age? There is not mention of age (strongly associated in univariable analysis) in the multivariable model (Figure 2). 

Thank you. As described in response to point 3 above, we have revised our modelling approach to be hypothesis driven, and adapted variable inclusion strategy to be based upon the causal pathways inferred by the directed acyclic graph. We hope that this principled approach provides strong rationale for the model specification approach taken.

We have now added additional text to describe the effect of index case age on household contact latent TB prevalence (lines 328-329), and have plotted new figures that we believe better convey the importance of household contact age on the risk of TST induration (Figure 1).

5. There is no discussion of the fact that proportion of day spent with index case showed a significant association with diameter induration but sleeping in the same bed with the index case was not. What are the possible speculations for these findings? 

In our revised model, neither proportion of day spent with index case nor sleeping in the same bed as the index case were identified as variables necessary to include in the final models. Thus the inconsistent results described above may have bene an artifact of the previous modelling approach.

6. Identifying predictors of larger diameter seems the main aim of the analysis. Nevertheless, message appear to be contradictory. Lines 241-243 implies that HIV-status of index case is indeed a predictor. In contrast in the Discussion it is stated that ‘In this study, we found that contacts of HIV-positive index cases had similar median posterior TST induration diameters to contacts of HIV-negative household contacts, even when stratified by study sites with markedly different epidemiological patterns of HIV and TB’. In general, there is a lack of standardisation in the presentation of results (HIV-status and age seem to be the only important predictor from reading the Conclusions of the abstract; HIV-status and duration of contacts with index case in the last paragraph of the Discussion). If the aim of the paper is direct public health efforts to specific target populations the identification of this population is rather confusing. 

Thank you for these helpful comments. We hope that the revised modelling approach described in response to points 1-5 above have addressed these concerns. In addition, we have made substantial revisions to the Discussion and Conclusion to provide greater clarity of the clinical and public health implications of the research findings.

7. Title of Table 1 should be. Characteristics of index cases (top) and household contacts (bottom) by Study site

Thank you, we have made this change.

8. Lines 224-227. From the introducing text a reader would expect to see the comparison of percentage of positivity in the 0-4 years old vs. >4 years old stratified by site, not the difference in proportion of positivity in the 0-4 years old by site alone. 

Thank you – we have revised this paragraph to add clarity.

9. Line 255. What was the p-value for this 3-way interaction? Wonder whether instead of describing the effect of age by HIV-status/study would have been more informative to give the differences according to HIV-status by age/study strata? 

In the revised models, we have removed analysis of the three-way interactions. Instead, we have presented fitted posterior distributions of TST positivity for index case age groups stratified by index case HIV status and microbiological status, study site, and household HIV status.

10. Abstract line 60. Data were collected prospectively as part of a household cluster-randomised trial in two South Africa provinces (Mangaung, Free State, and Capricorn, Limpopo). In intervention group households, TB contacts underwent HIV testing and tuberculin skin testing (TST).

I would rephrase making it clear from the start that this analysis only include the data from the intervention arm. Essentially is a cohort study using a single arm of a trial. This only becomes clear from reading the full paper. 

Thank you. We have added this to the Abstract

11. Page 16, four lines form the end. Typo mortality instead of morality

We have made this correction.

12. Page 19. what does BCG stand for? Spell out

BCG stands for Bacille Calmette–Guérin, the vaccine against tuberculosis. We have spelt this out.

13. Figure 1. Add a footnote to clarify that top of the figure is for cut off 5 mm and bottom for 10 mm? 

We have removed this figure in the revised version of the manuscript.

14. Figure 2. Are these mean differences in diameter or RR? RR of what? 

We have removed this figure in the revised version of the manuscript.

15. Figure 3 How can age of house contact be -1? Why is only the range [-1; +3] shown? 

We have removed this figure in the revised version of the manuscript.

RESPONSE TO REVIEWER 2’S COMMENTS

1. I commend the authors for the important study on the major public health problem in South Africa. Although study results are compelling, there are few methodological issues which need to be addressed/further explained in order to improve this manuscript.

We thank the Reviewer for these comments.

2. Page 18, paragraph 5, limitations: Authors have acknowledged that there were differences between individuals who received and did not TST in the study and these differences could have resulted in selection bias. I would suggest they use the data on those who did not receive TST to investigate how it could have influenced the results.

Thank you. In Supplemental Table 1, we have presented characteristics of participants, stratified by whether TST testing was done or not. We hope that this provides the clarity required. 

3. Page 6 line 151: Authors need to describe cut-offs of skin induration for LTBI under Tuberculin skin testing sub-section of Material and Methods section and which guidelines the cut-offs are based on. 

Thank you. We have added text to describe the two thresholds selected for analysis (≥5mm and ≥10mm) – Lines 205-208. In particular, we note that the updated 2018 latent TB guidelines emphasise that there is no gold-standard test for latent tuberculosis infection and that there is no consensus about optimal thresholds for a) use in predicting subsequent progression to active tuberculosis, or b) to inform treatment decisions, either for individual people, or groups of patients. Therefore, we elected to select two conventionally-used thresholds in our analysis: ≥5mm and ≥10mm.

---

## [Decision Letter · Decision Letter 1]

25 Feb 2020

PONE-D-19-29656R1

Prevalence and risk factors for latent tuberculosis infection among household contacts of index cases in two South African provinces: analysis of baseline data from a cluster-randomised trial

PLOS ONE

Dear Dr. MacPherson,

Thank you for submitting your manuscript to PLOS ONE. After careful consideration, we feel that it has merit but does not fully meet PLOS ONE’s publication criteria as it currently stands. Therefore, we invite you to submit a revised version of the manuscript that addresses the points raised during the review process.

Thank you for engaging so thoroughly and thoughtfully with the initial round of comments from the two reviewers. There are just a few remaining minor comments from reviewer #1. If these can be addressed I would be happy to accept the paper.

We would appreciate receiving your revised manuscript by Apr 10 2020 11:59PM. To enhance the reproducibility of your results, we recommend that if applicable you deposit your laboratory protocols in protocols.io, where a protocol can be assigned its own identifier (DOI) such that it can be cited independently in the future. For instructions see: http://journals.plos.org/plosone/s/submission-guidelines#loc-laboratory-protocols

We look forward to receiving your revised manuscript.

Kind regards,

Richard John Lessells, BSc, MBChB, MRCP, DTM&H, DipHIVMed, PhD

Academic Editor

PLOS ONE

Reviewers' comments:

Reviewer's Responses to Questions

**Comments to the Author**

1. If the authors have adequately addressed your comments raised in a previous round of review and you feel that this manuscript is now acceptable for publication, you may indicate that here to bypass the “Comments to the Author” section, enter your conflict of interest statement in the “Confidential to Editor” section, and submit your "Accept" recommendation.

Reviewer #1: (No Response)

Reviewer #2: All comments have been addressed

2. Is the manuscript technically sound, and do the data support the conclusions?

Reviewer #1: Partly

Reviewer #2: Yes

3. Has the statistical analysis been performed appropriately and rigorously? 

Reviewer #1: No

Reviewer #2: Yes

4. Have the authors made all data underlying the findings in their manuscript fully available?

Reviewer #1: Yes

Reviewer #2: Yes

5. Is the manuscript presented in an intelligible fashion and written in standard English?

Reviewer #1: Yes

Reviewer #2: Yes

6. Review Comments to the Author

Reviewer #1: This a much improved version of the analysis in which the authors have formulated a more specific hypothesis to be tested and constructed a directed acyclic graph (DAG) to specify the putative causal pathway between index TB case HIV status and household contact latent TB infection, accounting for mediating factors and confounders to identify the minimal sufficient adjustment set of covariates for estimating the direct causal effect. They also refined the statistical modelling by estimating a posterior distributions against more clinically meaningful outcomes (the proportion of household contacts with tuberculin skin test induration ≥5mm and ≥5mm).

I have a few extra requests of clarification.

1. From the described DAG there are only two identified confounders: index case age and province (labelled with a pink rectangular in the Figure). Therefore if the authors did use the minimal sufficient adjustment set of covariates for estimating the direct causal effect I would expect in Table 2 the models to be controlled for only these two factors and nothing else.

2. At page 20 of the revised Discussion the sentence that the results were obtained after controlling for both confounders and mediators is retained. This is worrying. For clarity to control for a mediator would be a mistake because the indirect effect is part of the causal effect that needs to be estimated so it does not need to be adjusted away.

3. It is ok to keep the DAG Figure as supplementary material but the text in the Methods need to be expanded to clarify the strategy used to construct the multivariable binomial regression. In other words, it has to be said that under the assumption described in the DAG age and province are the only two confounders and that controlling for these factors would remove all backdoor pathways of measured confounding. This appears later in results (lines 235-237) but the set of confounders listed (apart from index case age which is an ancestor of exposure and outcome) none of the others (study site, household contact sex and HIV status, sex, HIV status and microbiological TB status) seem to match the DAG.

4. I would also add a sentence in the limitations at page 22 of the Discussion to say that because only the intervention arm of the trial were used, unmeasured confounding is an issue. There are indeed latent variables described in the DAG which could be additional confounders that cannot be controlled for and that, more generally, the estimate can be interpreted as causal only under the assumption of a correctly specified DAG model.

Reviewer #2: (No Response)

7. PLOS authors have the option to publish the peer review history of their article (what does this mean?). If published, this will include your full peer review and any attached files.

Reviewer #1: No

Reviewer #2: No

---

## [Author Response · Author response to Decision Letter 1]

26 Feb 2020

Wednesday, 26 February 2020

Dear Prof Lessells,

Re: PONE-D-19-29656. Prevalence and risk factors for latent tuberculosis infection among household contacts of index cases in two South African provinces: analysis of baseline data from a cluster-randomised trial

I write on behalf of my co-authors to thank the reviewers for their careful second review, and appreciate the opportunity to respond with a revised submission. Please see a point-by-point response below.

We are particularly grateful for the Reviewers’ careful and consider suggestions for refinement to our analysis strategy, and again feel that they have provide substantial and positive input to this analysis, for which we are very grateful.

We look forward to your response.

Yours Sincerely,

Peter MacPherson

(On behalf of the authors)

RESPONSE TO REVIEWER 1’S COMMENTS

1. This a much improved version of the analysis in which the authors have formulated a more specific hypothesis to be tested and constructed a directed acyclic graph (DAG) to specify the putative causal pathway between index TB case HIV status and household contact latent TB infection, accounting for mediating factors and confounders to identify the minimal sufficient adjustment set of covariates for estimating the direct causal effect. They also refined the statistical modelling by estimating a posterior distributions against more clinically meaningful outcomes (the proportion of household contacts with tuberculin skin test induration ≥5mm and ≥5mm).

Thank you for these comments.

2. From the described DAG there are only two identified confounders: index case age and province (labelled with a pink rectangular in the Figure). Therefore if the authors did use the minimal sufficient adjustment set of covariates for estimating the direct causal effect I would expect in Table 2 the models to be controlled for only these two factors and nothing else.

Thank you for this comment and recommendation. We have revised the analysis to adjust for only index case age and province in the regression models now. Results are reporting in lines 274-302, and displayed visually in the new Figure 2. To allow readers to interpret data on TST positivity by other key characteristics (including age group of household contact), we have added a new Table 2 and Figure 1 reporting crude results.

3. At page 20 of the revised Discussion the sentence that the results were obtained after controlling for both confounders and mediators is retained. This is worrying. For clarity to control for a mediator would be a mistake because the indirect effect is part of the causal effect that needs to be estimated so it does not need to be adjusted away.

Thank you. We have corrected this sentence to note that only confounders were adjusted for in the analysis.

4. It is ok to keep the DAG Figure as supplementary material but the text in the Methods need to be expanded to clarify the strategy used to construct the multivariable binomial regression. In other words, it has to be said that under the assumption described in the DAG age and province are the only two confounders and that controlling for these factors would remove all backdoor pathways of measured confounding. This appears later in results (lines 235-237) but the set of confounders listed (apart from index case age which is an ancestor of exposure and outcome) none of the others (study site, household contact sex and HIV status, sex, HIV status and microbiological TB status) seem to match the DAG.

Thank you. We have corrected this description of the modelling strategy to reflect the adjustment for the minimally sufficient set of confounders (site and index case age) and to match the DAG.

5. I would also add a sentence in the limitations at page 22 of the Discussion to say that because only the intervention arm of the trial were used, unmeasured confounding is an issue. There are indeed latent variables described in the DAG which could be additional confounders that cannot be controlled for and that, more generally, the estimate can be interpreted as causal only under the assumption of a correctly specified DAG model.

We have added a sentence to the limitations to highlight the issue of potential unadjusted confounders, and the fact that the estimate can be interpreted as causal only under the assumption of a correctly specified DAG:

“Our estimate of the effect of index case HIV-positive status on TST positivity in household contacts can only be interpreted as causal under the assumption of a correctly specified DAG model. Moreover, unmeasured confounding may be an issue, particularly due to latent variables identified but uncontrolled for in the DAG.”

---

## [Editor Report · Decision Letter 2]

28 Feb 2020

Prevalence and risk factors for latent tuberculosis infection among household contacts of index cases in two South African provinces: analysis of baseline data from a cluster-randomised trial

PONE-D-19-29656R2

Dear Dr. MacPherson,

We are pleased to inform you that your manuscript has been judged scientifically suitable for publication and will be formally accepted for publication once it complies with all outstanding technical requirements.

With kind regards,

Richard John Lessells, BSc, MBChB, MRCP, DTM&H, DipHIVMed, PhD

Academic Editor

PLOS ONE
---

## [Editor Report · Acceptance letter]

5 Mar 2020

PONE-D-19-29656R2 

Prevalence and risk factors for latent tuberculosis infection among household contacts of index cases in two South African provinces: analysis of baseline data from a cluster-randomised trial 

Dear Dr. MacPherson:

I am pleased to inform you that your manuscript has been deemed suitable for publication in PLOS ONE. Congratulations! Your manuscript is now with our production department. 

With kind regards,

on behalf of

Dr. Richard John Lessells 

Academic Editor

PLOS ONE